# TRUNCATED PROXIMAL POLICY OPTIMIZATION

## ABSTRACT

Recently, test-time scaling Large Language Models (LLMs) have demonstrated exceptional reasoning capabilities across scientific and professional tasks by generating long chains-of-thought (CoT). As a crucial component for developing these reasoning models, reinforcement learning (RL), exemplified by Proximal Policy Optimization (PPO) and its variants, allows models to learn through trial and error. However, PPO can be time-consuming due to its inherent on-policy nature, which is further exacerbated by increasing response lengths. In this work, we propose Truncated Proximal Policy Optimization (T-PPO), a novel extension to PPO that improves training efficiency by streamlining policy update and length-restricted response generation. T-PPO mitigates the issue of low hardware utilization, an inherent drawback of fully synchronized long-generation procedures, where resources often sit idle during the waiting periods for complete rollouts. Our contributions are two-folds. First, we propose Extended Generalized Advantage Estimation (EGAE) for advantage estimation derived from incomplete responses while maintaining the integrity of policy learning. Second, we devise a computationally optimized mechanism that allows for the independent optimization of the policy and value models. By selectively filtering prompt and truncated tokens, this mechanism reduces redundant computations and accelerates the training process without sacrificing convergence performance. We demonstrate the effectiveness and efficacy of T-PPO on AIME 2024 with a 32B base model. The experimental results show that T-PPO improves the training efficiency of reasoning LLMs by up to 2.5× and outperforms its existing competitors.

## 1 INTRODUCTION

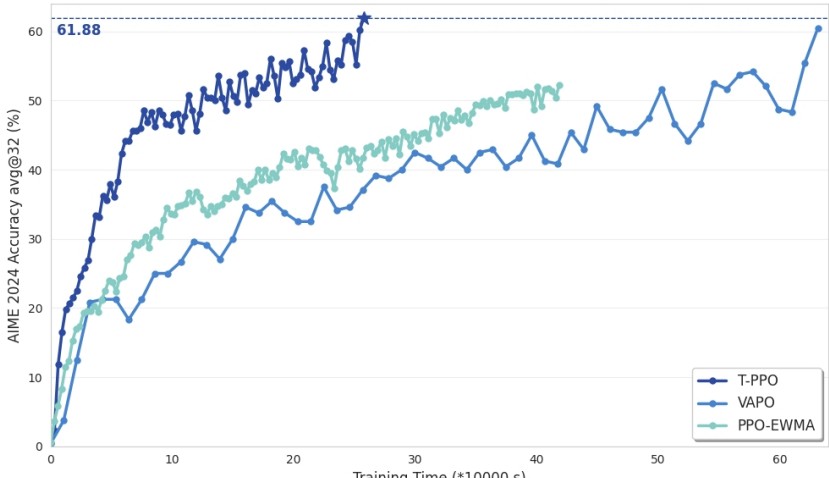

Figure 1: AIME 2024 scores of **T-PPO** on the Qwen2.5-32B base model, reduces training time by 60% compared to the previous state-of-the-art (SOTA) method. The values shown are pass@1 scores, averaged over 32 samples per question.

Recent advances in reasoning-oriented Large Language Models (LLMs), such as OpenAI's o1 (OpenAI, 2024), DeepSeek-R1 (Guo et al., 2025), and QwQ (Team, 2025), have demonstrated the state-of-the-art performance across complex domains including mathematical reasoning, programming, and agent-based tasks. These models leverage extended chain-of-thought (CoT) reasoning to improve inference quality, integrating backtracking and error-correction mechanisms that produce more structured and accurate outputs. This enhanced reasoning capability stems primarily from deep reinforcement learning (RL) techniques, through which LLMs learn to generate explicit, logically-sequenced reasoning steps prior to final answer production.

As the predominant RL approach for LLM refinement, Proximal Policy Optimization (PPO) (Schulman et al., 2017) maintains training stability through its clipped surrogate objective function. Despite its advantages, PPO's on-policy nature inherently restricts training efficiency, a limitation that becomes especially apparent when processing long CoT trajectories, often leading to substantial computational overhead and extended training durations. To address this issue, researchers have developed various off-policy PPO variants that are designed to enhance sample efficiency via trajectory reuse.

Specifically, Generalized Proximal Policy Optimization (GePPO) (Queeney et al., 2021) extends the guarantees of policy improvement to the off-policy setting. Off-Policy PPO (Meng et al., 2023) designs a clipped surrogate objective function that can utilize off-policy data and avoid excessively large policy updates. PPO-EWMA (Hilton et al., 2022) employs decoupled policy objectives and an exponentially weighted moving average (EWMA) for policy updates. KIMI K1.5 (Team et al., 2025) uses partial rollouts to improve training efficiency by reusing a large chunk of previous trajectories when sampling, thus avoiding the cost of regenerating new trajectories from scratch. Although off-policy methods are more training-efficient, they typically suffer from high variance in the policy gradient estimator, resulting in unstable training and degraded performance.

In this work, we present Truncated Proximal Policy Optimization (T-PPO), an enhanced on-policy reinforcement learning framework that significantly improves efficiency while maintaining or even enhancing reasoning performance. At the core of T-PPO is our Extended Generalized Advantage Estimation (EGAE) method, which enables progressive policy updates even before a trajectory is fully generated. Specifically, EGAE generalizes the conventional Generalized Advantage Estimation (GAE) (Schulman et al., 2015) to support policy optimization with partially generated responses. This decouples policy updates from response completion and significantly improves computational resource utilization. Furthermore, to ensure unbiased value estimation, we maintain the Monte Carlo training paradigm for value model updates, deferring these updates until full response generation is complete. This estimation relies exclusively on actual observed returns rather than estimated values, thereby eliminating approximation bias in the value function. This dual optimization strategy allows simultaneous, yet independent improvement of both policy and value models through selective token screening. In summary, our design yields three key advantages: (1) a truncated rollout strategy that enhances GPU throughput, (2) complete elimination of persistent bias in value function estimation, and (3) substantially improved policy update efficiency achieved through enhanced data utilization.

Our extensive experiments on the AIME 2024 benchmark demonstrate that T-PPO delivers significant efficiency gains without compromising model performance. Specifically, the algorithm exhibits robust convergence behavior through its sample-efficient learning mechanism, which consistently enhances policy optimization. These combined attributes enable T-PPO to achieve 62 pass@1 on the AIME'24 benchmark while demonstrating 2.5× higher training efficiency compared to state-of-the-art synchronization algorithms. Such performance characteristics significantly expand the practical deployment potential of T-PPO in real-world applications. Remarkably, these improvements are attained without introducing additional constraints or regularization beyond standard PPO. Apart from reducing the training cost, we sincerely hope that this method can bring more inspiration for delving into specialized expert models for professional domains.

## 2 PRELIMINARY

### 2.1 REINFORCEMENT LEARNING FRAMEWORK

Reinforcement Learning (RL) is a framework for sequential decision making. Typically, sentence generation can be formulated as a Markov decision process (MDP) represented by the tuple $\mathcal{M} = \{\mathcal{S}, \mathcal{A}, p, r, d_0, \gamma\}$. Here $\mathcal{S}$ is the state space, at each generation step $t$, the state $s_t = \{x, y_{1:t-1}\}$ is the concatenation of the input question $x$ and the output response generated so far $y_{1:t-1}$. $\mathcal{A}$ represents the action space and $p(s'|s, a) : \mathcal{S} \times \mathcal{A} \times \mathcal{S} \to [0, 1]$ is the transition probability distribution. In sentence generation, the transition probability is not needed because, given the current state and action, next state is deterministic. $r : \mathcal{S} \times \mathcal{A} \to \mathbb{R}$ is the scalar reward function with $r(s_t, a_t)$ for every intermediate time step $t$. Generally, the reward function $r(s_t, a_t)$ is defined for every state-action pair to provide feedback throughout the trajectory. In this work, we focus on the challenging case where rewards are non-zero only at the terminal timestep $T-1$, reflecting the correctness of the whole reasoning chain. Unlike dense-reward settings, this sparse-reward scenario naturally reduces to a bandit problem formulation. Note that our approach can be easily applied to simpler scenarios with process rewards. $d_0$ is the initial state distribution and $\gamma \in [0, 1]$ is a discount factor. The actions are taken from a probability distribution called policy $\pi$ given the current state $a_t \sim \pi(s_t)$. The goal is to choose a policy that maximizes the expected total discounted rewards $J(\pi) = \mathbb{E}_{\tau \sim \pi} \left[ \sum_{i=0}^{T-1} \gamma^i r(s_i, a_i) \right]$ where $\tau = (s_0, a_0, ..., s_t, a_t, ..., s_{T-1}, a_{T-1})$ is the trajectory generated by the LLM's interaction with environment $\mathcal{M}$ with $T$ being the length of the trajectory. Under a given policy $\pi$, the state-value function is defined as $V^\pi(s_t) = \mathbb{E}_{\tau \sim \pi}[G_t|s_t]$ where $G_t = \sum_{i=0}^{T-1-t} \gamma^i r(s_{t+i}, a_{t+i})$ is the discount return. Similarly, the state-action value function, i.e., Q-function, is defined as $Q^\pi(s_t, a_t) = \mathbb{E}_{\tau \sim \pi}[G_t|s_t, a_t]$, and the critical advantage function as $A^\pi(s_t, a_t) = Q^\pi(s_t, a_t) - V^\pi(s_t)$.

### 2.2 PROXIMAL POLICY OPTIMIZATION

PPO is a popular actor-critic reinforcement learning algorithm that has become a default baseline in LLMs. It optimizes LLMs by maximizing the clipped surrogate objective function

$$\mathcal{J}_{\text{PPO}}(\theta) = \mathbb{E}_{t, s_t, a_t \sim \pi_{\theta_{\text{old}}}} \left[ \min \left( \frac{\pi_\theta(a_t|s_t)}{\pi_{\theta_{\text{old}}}(a_t|s_t)} \hat{A}_t, \text{clip}\left(\frac{\pi_\theta(a_t|s_t)}{\pi_{\theta_{\text{old}}}(a_t|s_t)}, 1 - \epsilon_{\text{low}}, 1 + \epsilon_{\text{high}}\right) \hat{A}_t \right) \right] \quad (1)$$

, where $\pi_\theta$ and $\pi_{\text{old}}$ represent the current and previous policy respectively. $\hat{A}_t$ is an estimator of the advantage function, and $\epsilon_{\text{low}}$ and $\epsilon_{\text{high}}$ are hyperparameters that control the maximum deviation from the previous policy $\pi_{\theta_{\text{old}}}$. $\hat{A}_t$ is computed using generalized advantage estimation (GAE) (Schulman et al., 2015) based on rewards and a learned value function. The clipping objective of PPO restricts how drastically the updated policy distribution can diverge from the original policy. This moderation averts catastrophic shifts in language generation and preserves training stability. PPO limits the difference between consecutive policies by eliminating the incentive for the probability ratio $\frac{\pi_\theta(a_t|s_t)}{\pi_{\theta_{\text{old}}}(a_t|s_t)}$ to leave the clipping range $[1-\epsilon_{\text{low}}, 1+\epsilon_{\text{high}}]$, thus resulting in stable policy improvement throughout the learning process. However, it is well-known that high variance is a major issue in reinforcement learning, so often the number of samples must be large in order for the surrogate objective to be an accurate estimator of the true objective. Because these samples must be collected under the current policy between every policy update, PPO can be very inefficient.

### 2.3 GENERALIZED ADVANTAGE ESTIMATION

GAE provides a trade-off between bias and variance in the advantage estimation by combining multiple $n$-step advantage estimates through an exponentially weighted average controlled by the parameter $\lambda$. The advantage is computed as

$$\hat{A}_t = \delta_t + (\gamma\lambda)\delta_{t+1} + ... + (\gamma\lambda)^{T-t-1}\delta_{T-1} \quad (2)$$

, where

$$\delta_t = r_t + \gamma V(s_{t+1}) - V(s_t) \quad (3)$$

is the TD (temporal difference) residual, and $\gamma$ is the discount factor that determines how much future rewards are valued relative to immediate rewards. $\lambda$ is the GAE parameter that controls the

weighting of different multi-step estimates. GAE provides a way to control the bias-variance trade-off by adjusting $\lambda$, which allows us to tailor the advantage estimation to the specific problem.

## 2.4 VALUE FUNCTION ESTIMATION

In the PPO algorithm, the critic model, often referred to as the value function, estimates the expected returns for each observed state. This estimation achieves both variance reduction in policy gradients and generation of supplementary learning signals, which are essential to T-PPO's algorithmic design. A variety of different methods can be used to estimate the value function. When using a network to represent the value function, the simplest approach is to solve a nonlinear regression problem

$$V_{\phi,\text{CLIP}}(s_t) = \text{clip}\Big(V_\phi(s_t), V_{\phi_{\text{old}}}(s_t) - \xi_{\text{low}}, V_{\phi_{\text{old}}}(s_t) + \xi_{\text{high}}\Big) \tag{4}$$

$$\mathcal{J}_{\text{value}}(\phi) = \frac{1}{2}\mathbb{E}_{t,s_t,a_t \sim \pi_{\theta_{\text{old}}}}\Big[\max\big((V_\phi(s_t) - R_t)^2, (V_{\phi,\text{CLIP}}(s_t) - R_t)^2\big)\Big] \tag{5}$$

, where $V_\phi$ and $V_{\phi_{\text{old}}}$ are the current and previous value functions , respectively. $R_t = \sum_{i=0}^{T-1-t} \gamma^i r_{t+i}$ denotes the discounted return, and $t$ indexes over all timesteps in a batch of trajectories. The clipping operation enforces a constraint on the value function updates through hyperparameters $\xi_{\text{low}}$ and $\xi_{\text{high}}$, ensuring that the updated value function $V_\phi$ does not deviate significantly from $V_{\phi_{\text{old}}}$. This is sometimes called the Monte Carlo or TD(1) (Sutton et al., 1998) approach for estimating the value function. For reasoning tasks, we employ Monte Carlo estimation to maintain strictly unbiased state-value predictions, deliberately avoiding the use of GAE despite its variance-reduction benefits, as GAE introduces approximation bias through its temporal difference components.

## 3 T-PPO: TRUNCATED PROXIMAL POLICY OPTIMIZATION

To address the training inefficiency inherent in PPO, we propose Truncated Proximal Policy Optimization (T-PPO), a novel approach that enables policy optimization using incomplete trajectories. This section presents the technical framework of T-PPO through three key components:

- First, we introduce Extended Generalized Advantage Estimation (EGAE), a generalization of conventional GAE that accommodates partially generated responses. This innovation allows for:

  advantage computation from unfinished trajectories

  progressive policy updates during response generation

- Second, we detail our token-level optimization strategy, which features:

  selective filtering of training tokens

  independent yet simultaneous updates for policy and value models

- Finally, we present:

  the complete T-PPO algorithm implementation

  comprehensive analysis of training efficiency gains

## 3.1 EGAE: EXTENDED GENERALIZED ADVANTAGE ESTIMATION

This section focuses on producing an estimate $\hat{A}_t$ of the advantage function $A^\pi(s_t, a_t)$, when the entire trajectory is truncated. As discussed in the previous section, the GAE advantage estimator has a simple formula involving a discounted sum of Bellman residual terms. By introducing a parameter $\lambda$ that controls the influence of future reward on the advantage estimate, that is "forgotten" after $\approx 1/(1 - \lambda\gamma)$ timesteps, GAE provides a flexible bias-variance trade-off. As each TD residual term becomes more heavily discounted through a bootstrapping procedure, truncation of the trajectory has a negligible effect on the actions in the front position. In addition, our heuristic framework assumes that the generation of a single token does not significantly alter the state-value. That means for a truncated trajectory $\tau = (s_0, a_0, ..., s_{l-1}, a_{l-1})$ with truncation length $l$, we reasonably assume $V(s_l) = V(s_{l-1})$ for the next state that has not been generated yet. Under this assumption, the advantage estimate has the same format as in the non-truncated case (2) by simply replacing $T$ by $l$.

$$\hat{A}_t = \delta_t + (\gamma\lambda)\delta_{t+1} + ... + (\gamma\lambda)^{l-t-1}\delta_{l-1} \tag{6}$$

## 3.2 TOKEN FILTERING STRATEGY

PPO batches a group of prompts and waits all the prompts to complete their generation. In reasoning model training, the response length of each request varies greatly, resulting in GPU underutilization. Truncated Proximal Policy Optimization truncates the generation by a given maximum length, which we refer to as the window length $l$ in the following discussion to distinguish it from the actual maximum response length which may be generated by multiple rounds. This truncated rollout strategy effectively addresses the challenges associated with batching in LLMs training. If some sequences reach an ending condition, we remove these sequences in the next training step, while incomplete samples will be retained, and the total batch size of each training step is a constant. Figure 2 gives an example of this batching strategy of two consecutive steps.

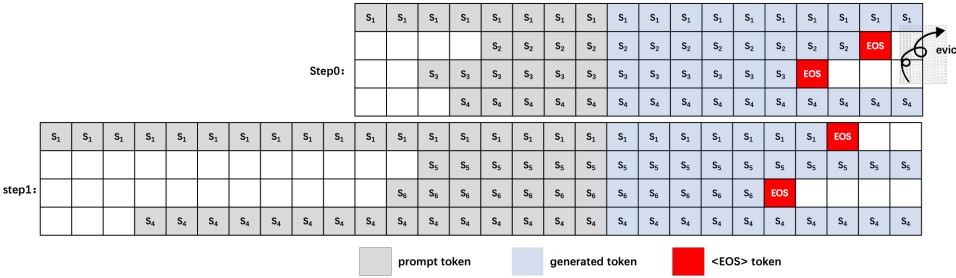

Figure 2: Successive batching strategy of T-PPO. Gray: prefill (computing input tokens), blue and red: decode (generating new response tokens). In step 0, sequences S2 and S3 emit an end-of-sequence token (red), so in step 1 we insert new prompts in their place (i.e. sequences S5 and S6), while the unfinished sequences continues in the next iteration. Each sequence finishes at different iterations.

As shown in Figure 3, taking training step 1 as an example, since the value model is trained in Monte Carlo mode, all generated tokens of finished sequences are used to train the value model. The policy model, in contrast, is trained on response tokens generated in the current training step, regardless of whether the sequence is complete or not. Additionally, since the advantage estimation of the latest generated tokens of unfinished sequences may be of high variance, we can exclude some latest generated tokens from policy model training.

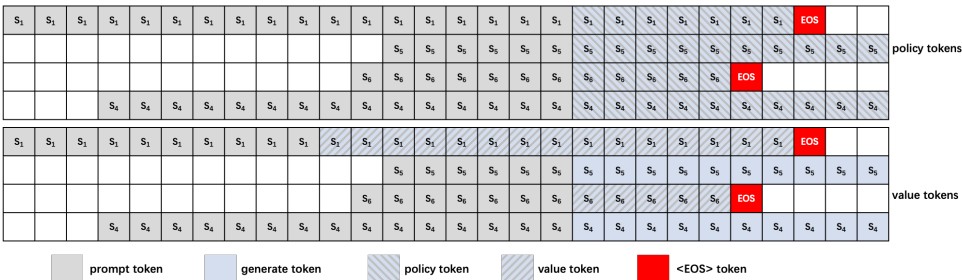

Figure 3: Plots showing one step (i.e. step 1) of the training tokens for policy model (top side) and value model (bottom side).

## 3.3 IMPLEMENTATION AND TRAINING EFFICIENCY ANALYSIS

T-PPO, which we detail in Algorithm 1, represents a principled approach to improving the training efficiency of PPO while retaining its approximate policy improvement guarantees.

As primary bottleneck of RL training lies in sample generation stage, due to the barrel effect, the walltime of sample generation phase is approximately proportional to maximum response length. When we truncate the response process, suppose that $L/l = k$ where $L$ is the actual maximum response length and $l$ is the window length, generation time saving is approximately $k$ times. In the training stage, from the perspective of each cross-turn response token, it is trained once each by the

---

**Algorithm 1** **T-PPO**: **T**runcated **P**roximal **P**olicy **O**ptimization

---

**Input** initial policy network parameters $\theta_0$, value function parameters $\phi_0$, task prompts $\mathcal{D}$, window length $l$
1: **for** step $j = 0,1,2...$ **do**
2:    Collect $K$ prompts from last step unfinished samples and replace finished samples with new prompts from $\mathcal{D}$
3:    Update the old policy model $\pi_{\theta_{old}} \leftarrow \pi_\theta$
4:    Collect trajectories $\{\tau_i\}_{i \in [0,K]}$ by running policy $\pi_{\theta_{old}}$ in the environment until $l$ time steps
5:    Compute rewards $\hat{R}_t$ for each finished trajectory
6:    Calculate advantage estimation $\hat{A}_{i,t}$ by EGAE (6) for each ($t$-th) policy tokens
7:    **for** minibatch = 0,1,2,... **do**
8:       Update the policy model $\pi_\theta$ by maximizing the T-PPO objective (1)
9:       Update the value function via gradient descent $\mathcal{J}_{\text{value}}(\phi)$ (5) on value tokens
10:    **end for**
11: **end for**
**Output** $\pi_\theta$

---

policy model and the value function, so the training time is also saved about $k$ times. Therefore, the end-to-end training efficiency ratio is about $k$ times for the same number of training steps.

## 4 EXPERIMENTS

In addition to the theoretical support for our algorithm presented in the previous section, we aimed to investigate the stability and training efficiency of T-PPO experimentally. In this section, we present detailed experimental results for T-PPO. We begin with an overview of the training configuration and datasets. Subsequently, we provide evaluation results and a comparison of training efficiency with several baseline methods. Finally, we investigate training dynamics, with particular focus on critical metrics such as response length evolution.

### 4.1 EXPERIMENTAL SETUP

#### 4.1.1 MODELS AND CONFIGURATIONS

In our experiments, we used Qwen-2.5-Base-32B as the initial checkpoint . The policy is trained with a constant learning rate of 1e-6, using the AdamW optimizer ($\beta = [0.9, 0.95]$) with weight decay 0.1, while the critic's learning rate was set as 2e-6. For fair comparisons, we applied hyperparameters similar to those of VAPO: a batch size of 512 prompts, sampling 16 times per prompt, and setting the minibatch size to 512. The value network was initialized using a reward model, with the GAE $\lambda$ set to 0.95 and $\gamma$ set to 1.0. Token-level loss was used, and we set the clipping parameters $\epsilon_{\text{low}} = 0.2$ and $\epsilon_{\text{high}} = 0.28$ for the policy and $\xi_{\text{low}} = 0.5$, $\xi_{\text{high}} = 0.6$ for the value function. For evaluation on AIME, we repeat the evaluation set 32 times and report avg@32 for the stability of the results. The inference hyperparameters of evaluation were set to temperature 1.0 and topp 0.7. In addition, given the significant distribution shift between the reasoning model and the base model, we removed the KL divergence from the loss function to encourage exploration. We set the maximum response length to 24k while window length to 8k in T-PPO.

#### 4.1.2 DATASET DESCRIPTION

To fully demonstrate the effectiveness and efficiency of our proposed algorithm, we conducted experiments using the American Invitational Mathematics Examination (AIME) as a representative benchmark for reasoning problems. The AIME often requires a long chain of thought to solve. The test set comprises AIME problems from the last year. The training set, DAPO-Math-17K (Yu et al., 2025), consists of questions from all past AIME competitions, supplemented by some artificially constructed difficult math problems. We implemented the verification-based reward function using

Table 1: Results of different algorithms on AIME

| Model | AIME24$_{avg@32}$ |
|---|---|
| DeepSeek-R1-Zero-Qwen-32B | 47 |
| DAPO Yu et al. (2025) | 50 |
| VAPO Yue et al. (2025) | 60 |
| GePPO Queeney et al. (2021) | 50 |
| PPO-EWMA Hilton et al. (2022) | 52 |
| **T-PPO** | **62** |

Math-Verify, with the following minimalistic rule:

$$R(x, y) = \begin{cases} 1 & \text{if } y \text{ contains the correct final answer to } x \\ 0 & \text{otherwise} \end{cases} \tag{7}$$

We run different RL algorithms on questions sampled from the training dataset and compare the vanilla PPO, representative asynchronous PPO-EWMA with the proposed T-PPO.

## 4.2 EXPERIMENTAL RESULTS

### 4.2.1 MAIN RESULTS

The main results comparing T-PPO with baseline methods across the AIME dataset are presented in Figure 1 and Table 1. Our approach ultimately achieves 61.88 pass@1 on AIME 24, surpassing the performance of DeepSeek-R1-Zero-Qwen-32B and SOTA async PPO algorithm available, matching PPO with 20k response length with 60% wall-clock time reduction on the AIME24 benchmark. T-PPO exhibits high training stability and better training efficiency, making it the preferred choice in this setting.

### 4.2.2 TRAINING EFFICIENCY

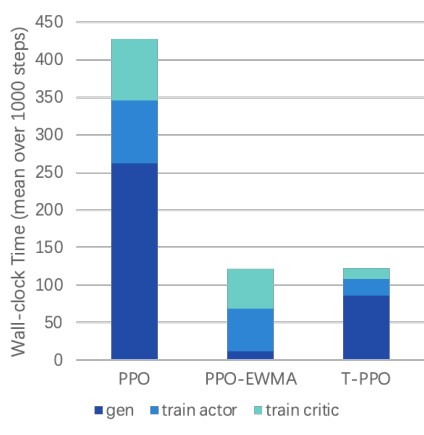

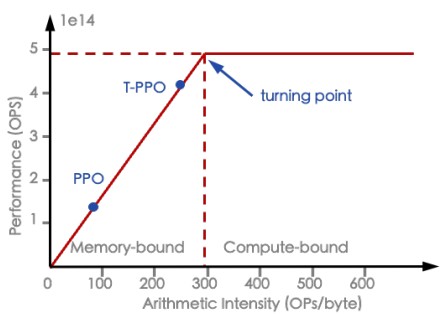

Figure 4: Algorithm comparison in terms of time efficiency on the AIME benchmark. Each boxplot is drawn based on the execution mean of 1000 steps.

Figure 5: Demonstration of the Roofline model of Nvidia H800 GPU. The computation is in BF16.

To further understand the efficiency improvement of T-PPO, we show its RL iteration breakdown and compare it with other algorithms. We divide one RL iteration into three main parts: the generation stage (gen), the policy training stage (train actor) and the value training stage (train critic). Figure 4 compares the time efficiency of different algorithms. The average wall-clock time consumption per 1000 steps of T-PPO is comparable to that of PPO-EWMA and much lower than that of vanilla PPO

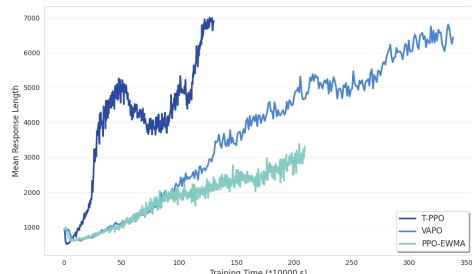

Figure 6: The metric curves of response length.

algorithm. In addition, although T-PPO and PPO-EWMA have similar per-step wall-clock time, T-PPO has significantly fewer convergence steps than PPO-EWMA (i.e., 6720 vs. 11200 steps each). Thus, both vanilla PPO and asynchronous PPO require much more total run time to converge, which may impede its applications to solving real-world problems.

Beyond temporal efficiency comparisons, our Roofline analysis Figure 5 - as reflected in the computational intensity profiles - provides more profound architectural insights into the system's performance characteristics. T-PPO demonstrates a computational intensity of 249 operations/byte in policy rollout, significantly higher than PPO's 84 operations/byte, positioning it closer to the arithmetic peak on the Roofline curve. This indicates that T-PPO better utilizes compute resources by reducing memory-bound bottlenecks through optimized GPU utilization in the generation stage.

### 4.2.3 TRAINING DYNAMICS

The length of generated responses serves as an indicator of training stability and model capability, as illustrated in Figure 6. Our analysis reveals a characteristic fluctuation pattern in which response length initially increases, undergoes a temporary decline, then recovers before eventually stabilizing. This non-monotonic trajectory suggests that the model continuously refines its reasoning methodology throughout the learning process. Importantly, the final stabilized responses length surpasses the vanilla PPO, demonstrating that our method preserves (and potentially enhances) the reasoning model's length scaling capacity. The initial length expansion provides greater exploration space for complex reasoning behaviors, consistent with the observed "emergence" of lengthy chain-of-thought (CoT) through RL training (Zeng et al.; Hu et al., 2025). The eventual recovery and surpassing of initial lengths indicates the model's successful navigation through this transitional phase to achieve superior reasoning capacity.

## 5 CONCLUSION

In this work, we introduce Truncated Proximal Policy Optimization (T-PPO), a novel extension of PPO that incorporates a successive batching strategy to enhance the training efficiency. By leveraging an innovative extended generalized advantage estimation in conjunction with computationally efficient mechanisms to optimize the policy and the value models respectively, it achieves up to 2.5× training speedup with competitive final performance. Our detailed experiments and empirical insights provide practical guidance and valuable experience for future research on efficient large-scale reinforcement learning. Further exploration of truncated or more advanced RL methods is recommended.

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

## A  THE USE OF LLMs

This paper utilized Large Language Models to assist with text polishing and grammar checking. All AI-generated content has been reviewed and verified by the human author, who takes full responsibility for the paper's content.

