# OpenReview forum: "Truncated Proximal Policy Optimization"
_ICLR.cc/2026/Conference — Submitted to ICLR 2026_

### Official Review · Reviewer_64VD · 2025-10-17

**Soundness:** 2
**Presentation:** 1
**Contribution:** 1
**Rating:** 0
**Confidence:** 5

**Summary:**

This paper proposes Truncated Proximal Policy Optimization (T-PPO), an algorithm aimed at improving the training efficiency of LLM on reasoning tasks. The core idea is to mitigate the hardware underutilization caused by PPO's on-policy requirement to wait for full sequence generation. T-PPO introduces two main components: (1) Extended Generalized Advantage Estimation (EGAE), a method to compute advantage estimates from incomplete trajectories by truncating the advantage sum, and (2) a decoupled optimization strategy where the policy is updated frequently on partial data, while the value function is updated less frequently on completed trajectories. The authors test their method on the AIME 2024, showing speedup and performance improvement compared to baselines.

**Strengths:**

1.	The paper studies a critical and widely recognized bottleneck in RL for LLMs: the inefficiency of on-policy sampling.

2.	The paper reports good improvement and speedup over baselines on math reasoning tasks.

**Weaknesses:**

1.	The prposed EGAE appears to be the standard truncated-trajectory GAE which is proposed in the original PPO paper [1]. Truncated fixed-length trajectory segments is long used for GAE in PPO. This is not novel.

2.	The evaluation has significant weakness. The reliance on a single benchmark and a single model prevents any claims of general applicability. Although the paper claims faster convergence, but only the convergence steps for the proposed T-PPO and PPO-EWMA are reported, with other baseline methods’ statistics missing.

3.	The paper's proposed T-PPO seems to be more related to asynchronous reinforcement learning methods for training LLMs, which also aims to decouple data collection and model updates to improve wall-clock efficiency [2]. The paper should add discussions and compare with these methods.

[1] Proximal Policy Optimization Algorithms

[2] Beyond Ten Turns: Unlocking Long-Horizon Agentic Search with Large-Scale Asynchronous RL

**Questions:**

Please see Weaknesses Section

---

### Official Review · Reviewer_MPdS · 2025-10-22

**Soundness:** 2
**Presentation:** 3
**Contribution:** 2
**Rating:** 2
**Confidence:** 4

**Summary:**

This paper proposes an improved policy-gradient reinforcement learning method based on Proximal Policy Optimization (PPO), termed Truncated PPO (T-PPO). The primary contribution lies in the training system design, which substantially improves GPU utilization during training.

However, the proposed Extended Generalized Advantage Estimation (EGAE) appears flawed. The value-function update can exhibit highly variable effective batch sizes, potentially leading to unstable updates.

The experimental evaluation is also quite limited: only one model and one dataset are considered. The paper appears to report a single training run, with no sensitivity analysis over random seeds, datasets, or model architectures. Key ablations (e.g., window length) are missing.

**Strengths:**

1. The paper is clearly written and well organized, making the method easy to follow.
2. The proposed approach appears computationally efficient for the chosen model and dataset.

**Weaknesses:**

1. The paper lacks a Related Work section. The authors should review recent advances in related areas.
2. The value function estimates the expected return under the current policy; the two should be tightly coupled. The current design partially decouples value training from policy updates.
3. The experiments are too limited: only one model and one dataset are used, seemingly with a single training run. There is no robustness analysis across seeds, datasets, or models.
4. Ablation studies on key components—such as the truncation window length—are missing.

**Questions:**

1. The proposed "Extended Generalized Advantage Estimation (EGAE)" is flawed from my perspective. Before training start, value function should always give "0" for any trajectory. The KL penalty will not work as $\pi_\theta=\pi_{ref}$. In such case, in GAE, $\delta_t$ should be 0 for all $t\in\{1,2,3...,T-1\}$, but for last token, $\delta_T=r_T$. If the returns are truncated by a finite window, the estimated advantages may be substantially biased, potentially causing instability in the initial training stage. How is this addressed?
2. Did the authors warm up the value function before commencing T-PPO training? If so, how many steps were used? If not, how do you mitigate the issue raised in Q1?
3. In Figure 1, the reward curves increase monotonically and do not appear to converge. What happens if training continues for more steps?
4. The response length in Figure 6 appears unstable. In my experience, such instability can lead to large variance across datasets or random seeds. Could the authors provide additional results to assess robustness?
5. During policy updates, all generated tokens are included. However, the value-function update rule seems to induce large variability in the total number of tokens per batch. Since losses are averaged over the batch, could this introduce instability or bias the updates toward particular tokens? Have the authors considered or controlled for this effect?

---

### Official Review · Reviewer_SM7P · 2025-10-31

**Soundness:** 4
**Presentation:** 4
**Contribution:** 2
**Rating:** 4
**Confidence:** 4

**Summary:**

This paper proposes TPPO, an improved asynchronous version of PPO designed to enhance computational efficiency. TPPO achieves more than 2.5× faster computation while maintaining nearly the same performance as standard PPO.

**Strengths:**

This is a method that the research community needs. Because Chain-of-Thought models produce very long token sequences, training them is practically infeasible for most researchers outside major companies. Although the proposed method is limited to PPO, a technique that accelerates training for long output sequences has significant practical value.

The method description is clear and easy to follow.

The experiments convincingly support the authors’ claims.

**Weaknesses:**

Limited experimental comparisons.
The paper reports experiments on only one dataset. The core assumption of TPPO,
V(s_l) = V(s_{l-1}), is a strong approximation whose theoretical error bound is not proven. Therefore, the authors should conduct experiments on multiple datasets to demonstrate that the approximation holds in general. In particular, it is unclear whether TPPO performs well on shorter datasets where the value function changes rapidly, or on non-mathematical domains. Based on the presented results alone, it is difficult to confirm the generality of TPPO.

Insufficient ablation studies.
The paper presents results only on the Qwen2.5-32B model. While intuitively the method should work on other models as well, the paper should still provide empirical evidence—such as how much the speed improvement varies with model size. Moreover, there is even one unused page (the paper uses 8 out of 9 pages), so additional experiments could easily be included. More thorough ablation studies would greatly strengthen the paper’s credibility. Specifically, the effects of the Dual Update Mechanism and Token Filtering Strategy are underexplored. Since the Dual Update Mechanism could in principle be used even without TPPO, its independent contribution to performance should be analyzed.

Limited novelty.
The proposed Token Filtering Strategy and Dual Update Mechanism seem more like engineering optimizations than conceptual innovations. Only the EGAE component appears to address a new issue in PPO, yet even that looks like a minor variant of GAE. While the paper has some engineering novelty, it lacks deeper insight into reinforcement learning or AI theory.

**Questions:**

The Dual Update Mechanism appears applicable to standard PPO as well. What would be the results if applied to vanilla PPO? Would performance or speed differ?

How would the speedup of TPPO change with smaller models, e.g., a 4B model instead of 32B? Would the performance gap between PPO and TPPO still be significant?

---

### Official Review · Reviewer_dNEk · 2025-11-01

**Soundness:** 3
**Presentation:** 3
**Contribution:** 2
**Rating:** 4
**Confidence:** 5

**Summary:**

This paper aims to extend on-policy RL methods such as PPO to the off-policy or partial-rollout setting. The authors propose T-PPO, which modifies the GAE formulation to handle truncated trajectories, enabling effective learning when only partial rollouts are available.

**Strengths:**

The target problem is important, as partial rollouts naturally arise in large-scale RL for LLMs. The presentation is clear, and the reported experimental results show promising improvements.

**Weaknesses:**

The method is incremental. The proposed EGAE differs little from standard GAE, and the overall algorithm resembles prior partial-rollout implementations such as those used in Kimi. The paper does not clearly justify what is new or why the proposed adjustment yields improvement.

In addition, the analysis is limited. There is no theoretical or empirical discussion of bias introduced by truncation, nor comparison with other off-policy corrections such as partial rollout with importance sampling, which would be the most direct baseline.

**Questions:**

Is the token filtering step equivalent to partial rollouts rather than an actual filtering mechanism?

---

### Meta-Review · Area_Chair_SouD · 2025-12-26

**Summary:**

This paper targets the time-efficiency of Proximal Policy Optimization (PPO). The authors introduce the Truncated Proximal Policy Optimization (T-PPO) method to enhance the training efficiency of PPO. In terms of the reviewers’ concerns, all reviewers gave negative scores to this paper. Specifically, the following concerns inform me of the rejection decision for this paper.

- All the reviewers highlight the limited experiments and the ablation studies. For example, the authors only use one model and one test dataset in the experiments. In addition, there is no ablation on the key component of T-PPO, such as the truncation window length.

- The reviewer MPdS points out that this paper does not have the related work section, which I think is a critical error for the integrity of a scientific paper.

- Reviewers dNEk, SM7P, and 64VD raise concerns regarding the novelty of the proposed method. The proposed method seems to be a minor variant of existing methods.

Overall, I agree with the reviewers’ concerns. I think this paper lacks the necessary experiments and structural integrity. In addition, the authors did not post any response to address the reviewers’ concerns. Due to the above reasons, I would like to recommend the rejection. I suggest that authors perform more experiments and ablation analysis. In addition, authors should incorporate a related work section in the main text to keep the integrity of the scientific paper.

**Reviewer Concerns:**

Since the authors did not respond to any reviews during the discussion period, I think all the reviewer’s concerns are still outstanding. For example, the concerns of experiments and ablation studies, the novelty concerns, and so on.

**Reviewer Scores:**

Since the authors did not respond to any reviews, I think none of the reviewers would have changed their scores if they had the opportunity to fully participate in the discussion.

---

### Decision · Program_Chairs · 2026-01-26

Reject